# The Analysis of the Nutritional Status and Dietary Habits among Children Aged 6–10 Years Old Attending Primary Schools in Poland

**DOI:** 10.3390/ijerph19020953

**Published:** 2022-01-15

**Authors:** Magdalena Potempa-Jeziorowska, Paweł Jonczyk, Elżbieta Świętochowska, Marek Kucharzewski

**Affiliations:** 1Chair and Department of Descriptive and Topographic Anatomy, Faculty of Medical Sciences in Zabrze, Medical University of Silesia, 40-055 Katowice, Poland; pawel_jonczyk@o2.pl (P.J.); kucharzewskimarek@poczta.onet.pl (M.K.); 2Department of Medical and Molecular Biology, Faculty of Medical Sciences in Zabrze, Medical University of Silesia, 40-055 Katowice, Poland; eswietochowska@sum.edu.pl

**Keywords:** children, dietary habits, obesity, overweight

## Abstract

A high prevalence of obesity among children is influenced by serious implications. Obesity mainly results from behavioral factors, such as improper dietary habits. This study aims to evaluate the nutritional status and dietary habits of children aged 6–10 (*n* = 908) attending primary schools in Poland, Europe. The research tool was a questionnaire that was completed by one of the children’s parents. A statistical analysis was made using statistical software. The value of *p* = 0.05 was considered to be statistically significant. A total of 74.7% of children surveyed have a normal body mass. As many as 91.7% and 76.6% of children, respectively, eat a first and second breakfast daily. Nearly half of parents (48.9%) state that their child consumes milk or other dairy products daily. A total of 74.3% of children drink water daily. A total of 27.6% eats fish less frequently than once a week. A total of 7.6% of children eat fish several times a week. As many as 20.6% of the respondents state that their child eats brown bread several times a week, whereas 19.9% state that their child never eats brown bread. A total of 55.1% of children eat fruits and/or vegetables daily. A total of 14.1% of children surveyed consume sweets daily. The study revealed a positive correlation between BMI and the frequency of mineral water consumption (*p* = 0.013) in 9 y.o. girls. It was also revealed that the number of consumed fruit/vegetables increases with the BMI value among 10 y.o. boys (*p* = 0.044). Conclusions: The dietary habits of the investigated children are still improper. There is a great need for education on this issue, but family involvement is also required.

## 1. Introduction

The increasing prevalence of overweight and obesity among children is one of the most serious public health challenges of the 21st century. Obesity has become a multifactorial disease influenced by serious medical, personal, environmental, genetic and social factors. Over the past 40 years, the global prevalence of obesity in children has increased to more than 5% [1]. In 1975, 0.7% of girls were diagnosed with obesity. In 2016, this percentage increased to almost 6%. When it comes to boys, this percentage increased from 1% in 1975 to approximately 8% in 2016. Looking at the numbers, 50 million girls and 74 million boys worldwide were obese in 2016 [2]. According to the European studies commissioned by the WHO in 2015–2017, the median prevalence of overweight and obesity was 29% in boys and 27% in girls aged 7–9 years. Obesity was diagnosed in 13% of boys and 9% of girls. In Poland, the prevalence of excessive body mass disorders is in the middle range across other European countries, or has even slightly higher than average values [3]. On the contrary, the prevalence of underweight decreased from 9.2% in 1975 to 8.4% in 2016 among girls and from 15% in 1975 to 12.4% in 2016 among boys. In 2016, 75 million girls and 117 million boys worldwide were moderately or severely underweight [2]. Excess body weight is associated with lifelong health consequences. Children are at a greater risk of many health disorders that used to be almost exclusively typical of adulthood, such as metabolic syndrome. Currently, pediatricians have to treat metabolic consequences of obesity, such as hypertension, dyslipidemia, impaired fasting glucose and insulin resistance. These disorders, along with obesity, are the main components of metabolic syndrome (MetS). There are many diagnostic criteria for MetS; however, none of them have been standardized. Nevertheless, obesity remains a key component of MetS [4]. The vast part of obesity etiology belongs to simple obesity, which results from a long-term positive energy balance. Improper dietary habits and a sedentary lifestyle promote this condition. It has been proven that, up to the age of 10, the eating habits that are being created are of the greatest importance. This is why the age group of 6–10 is very crucial for proper nutrition in the following childhood years.

This study aims to evaluate the nutritional status and assess the dietary habits of children aged 6–10 attending selected primary schools in Poland, Europe.

## 2. Materials and Methods

### 2.1. Study Design

Descriptive population research study that focuses on assessment of nutritional status and dietary habits in children was evaluated. The study participants were children aged 6–10 attending selected primary schools in Poland (Europe). The research tool was a questionnaire completed by one of the children’s parents. This is the article, which contains data obtained from the first and second part of the survey.

The survey was conducted according to the principles of the Declaration of Helsinki. Participation in the study was voluntary and anonymous. The participants and their legal guardians provided written informed consent for participation in the study.

### 2.2. Settings

The questionnaire was originally designed by authors and its content was based on the *Nutritional Standards for the Polish population* (last published edition: 2017) developed by the National Food and Nutrition Institute [5]. The survey questions were divided into 4 sets. The first set of questions included information concerning self-reported anthropometric data of children (weight, height) and basic sociodemographic information. The second set of questions focused on children’s dietary habits, whereas the third one included questions concerning the level of children’s physical activity. The last set of questions was dedicated to the parents of children and it aimed to assess the level of their knowledge about proper nutrition guidelines.

The questionnaires were completed by parents during school meetings or at home after they were instructed by the authors of the paper. Receipt of surveys was during school meetings by the authors or by the child to the teacher during the next school day. Then, authors received all of the returned questionnaires together. Only the questionnaires with the first part fully completed were included in the analysis. This article published obtained data from the second part of the study.

### 2.3. Data Collection

From March 2019 to June 2019, the authors distributed 5000 questionnaires to 38 schools in Poland. Geographical distribution of questionnaires was 2 primary schools in every Polish region (number of regions in Poland—16), beside Silesian region, where questionnaires were distributed to 8 schools. Minimum number of students attending school was 150. The schools taking part were those whose headteachers gave their consent to the survey being conducted.

### 2.4. Statistical Treatment

The results were obtained using the professional software IBM SPSS Statistics version 25 (IBM Polska sp z o.o., Warsaw, Poland). The comparative analysis was performed using the parametric Student’s *t*-test for independent samples and the non-parametric Mann–Whitney *U* test and the Kruskal–Wallis test. The correlation analysis was performed using Pearson’s correlation (*r*) and Spearman’s correlation (*rho*). In this paper, the value of *p* = 0.05 is statistically significant.

## 3. Results

### 3.1. Participants

The authors received 1010 questionnaires back. The response rate was 20.2%.

This study involved 908 children, with a similar distribution by sex (*n* = 419 boys; *n* = 450 girls). The age range of the children was 6–10 years. Boys and girls aged 8 were the largest age group. Half of the studied children (*n* = 454) lived in a city (>100,000 inhabitants). In contrast, 14% of the children (*n* = 130) lived in a town (<100,000 inhabitants) and 36% (*n* = 324) in rural areas.

### 3.2. Nutritional Status

Based on the anthropometric data obtained, the BMI was calculated. To evaluate the nutritional status, the BMI value was marked on standardized BMI percentile charts according to the sex and age of the children surveyed. In this study, the BMI percentile charts developed by the WHO in 2007 for children aged 5–19 were used as a reference for the cut-off criteria for body mass disorders. A value of BMI between the 5–85th percentiles was assessed as a normal body mass, and the range between the 86–95th percentiles corresponds to overweight. A BMI at a value above the 95th percentile indicates obesity. In contrast, a BMI under the 5th percentile means undernutrition [6].

According to the results of this study, nearly ¾ (74.7%) of children surveyed have a normal BMI. Almost one child in five (17.1%) has excess body weight (overweight or obesity). Obesity is present in 6.8% of all children surveyed, whereas overweight is present in 10.3%. Underweight is present in 8.2% of all children surveyed. Table 1 shows the sex-specific distribution of the nutritional status of the studied group [7].

### 3.3. Dietary Habits

The dietary habits section of the questionnaire included questions concerning both quantitative and qualitative dietary habits. The following statements were used in the questionnaire to describe the frequency: everyday; several times a week (meaning three or four times); once a week; less than once a week; never.

The obtained results reveal that the majority of children surveyed eat a first breakfast daily (91.7%) before going to school. A total of 6.4% of respondents state that their child eats a first breakfast several times a week. A first breakfast constitutes the main meal at the beginning of the day. Regarding a second breakfast (meal eaten after breakfast, but before lunch), more than ¾ of respondents (76.6%) declare that their child eats a second breakfast daily at school. The survey results also reveal that 16.7% of children eat a second breakfast several times a week, whereas 1% of children eat a second breakfast once a week (0.2%) or less than once a week (0.8%). A similar proportion of children never eat a second breakfast (1.1%). The vast majority of children eat dinner daily (98.2%). In the questionnaire, there was also a question about supper frequency consumption as the last meal in a daily routine. The distribution of answers is as follows: 94.7% of children eat supper daily, and fewer than 5% of children (3.8%) eat this meal several times a week (Table 2).

The obtained results revealed no statistically significant correlation between boys and girls’ BMI and the frequency of eating the first and second breakfast and supper (*p* = 0.364; 0.936; 0.418, respectively). Table 3 shows obtained correlations.

The questionnaire included questions referring to the frequency of consumption of some food products. Nearly half of parents (48.9%) state that their child consumes milk or other dairy products daily. Slightly fewer respondents (38.4%) claim that their child consumes milk several times a week. The obtained results reveal that nearly 10% of children surveyed drink milk once a week (4.8%), less than once a week (2.8%) or never drink milk (1.9%). Mineral water was the most frequently consumed product. Nearly ¾ of children drink water daily, 15.5% drink water a few times a week and approximately 5.2% of children surveyed drink water once a week or less than once a week. Above half (52.1%) of investigated children eat fish products once a week. One child in four (27.6%) eats fish less frequently than once a week. Not more than 10% (7.6%) of children surveyed eat fish several times a week. When it comes to the consumption of bread made with wholemeal flour, approximately one in five parents stated that their child eats this product several times a week (20.6%) or never eats this product (19.9%). Most children, i.e., almost one third (28.4%), consume this kind of bread less frequently than once a week. Only every tenth child eats this product daily. When it comes to meat consumption, the obtained results show that most children (72.9%) consume meat several times a week. Nearly one in five children (17.1%) eats meat daily. Fewer than 1% of children do not eat meat at all. The obtained results regarding the frequency of the daily consumption of fruit and vegetables are as follows. More than half (55.1%) of children surveyed eat fruits and/or vegetables daily. Slightly fewer respondents (38.1%) stated that their child consumes such products several times a week. Approximately 5% of children eat fruit and/or vegetables once a week or even less frequently. Table 4 illustrates all the obtained data concerning the frequency of the consumption of milk, water, fish, brown bread, meat and vegetables/fruits.

The questionnaire also included a question concerning the frequency of beverage (which means non-alcoholic sugary drinks) consumption. The responses reveal that approximately 1/3 of studied children (35.9%) consume such products several times a week. Almost one in five children drink beverages once a week (18%) or even less than once a week (19.6%), and, 15% of children drink beverages daily. However, an important fact is that 8.5% of respondents left this question unanswered. When it comes to the frequency of the consumption of sweets and salty snacks, the responses indicate that fewer than half of studied children (43.6%) consume these products several times a week. Almost 1/5 children consume sweets once a week (18.3%) or even less frequently (17.5%). Nevertheless, 14.1% of children surveyed consume such products daily. It is significant to note that this question was left unanswered by 5.3% of the respondents (Table 4).

The obtained results showed that there was not any correlation found between the BMI of children aged 7 and 8 and the frequency of yjr consumption of milk and other dairy products, water, fish, wholemeal flour bread, meat and fruit/vegetables, regardless of sex. No correlations were also found among the 9 y.o. boys group. However, there was a statistically significant positive correlation between BMI and the frequency of mineral water consumption (*p* = 0.013) in 9 y.o. girls. There was also a negative correlation between BMI and the frequency of the consumption of wholemeal flour bread (*p* = 0.008) and meat (*p* = 0.013) in 10 y.o. girls. There was a statistically significant positive correlation between BMI and the frequency of fruits/vegetables consumption (*p* = 0.044) in 10 y.o. boys. The number of consumed fruits/vegetables increases with the BMI value. There was no correlation between the BMI of all studied children and the frequency of the consumption of beverages and sweets/salty snacks daily. Table 5 show all correlation results obtained.

The dietary habits section of the questionnaire also included a question concerning the number of snacks consumed per day. According to the results obtained, most children (41.0%) eat two snacks daily. Approximately one in five children consumes one (17.8%) or three snacks (22.1%) a day. Interestingly, approximately 11% of children surveyed eat four or even more snacks per day. When it comes to the frequency of consuming products in fast food bars, 68.3% of the respondents stated that their child eats meals in fast food bars less than once a month. More than one in five children (27.1%) consumes fast food products less than once a week. Nevertheless, 3.2% of children eat meals in fast food bars once a week.

The correlation analysis reveals that the BMI value of the studied children correlates neither with the number of snacks consumed per day nor with the frequency of consuming products in fast food bars across all age groups of children (Table 6).

The last question included in the dietary habits section of the questionnaire concerned the most popular product purchased in the school shop. In this question, there were a few options to choose: (a) my child does not buy at school shop; (b) “something sweet” (e.g., wafers, buns, candies, chocolate); (c) water; (d) beverages (which means non-alcoholic sugary drinks); (e) salty snacks; (f) fruits/vegetables; (g) other products. Based on the obtained results, approximately half of the children surveyed (46.1%) do not buy anything in the school shop, whereas 15.5% of children buy “something sweet” and 6.3% consume salty snacks from the school shop. A small number of children buy water (4.5%). A negligible proportion of children admit to buying beverages (1.5%) and fruits/vegetables (0.9%) (Figure 1).

The results of the correlation analysis indicate that the highest percentage of children, regardless of their BMI value, do not buy anything from the school shop. The only exception is overweight girls, who most frequently buy “something sweet”. The most popular product that was purchased in the school shop by overweight and obese boys and by girls with a normal BMI was also “something sweet”. In the case of overweight and obese girls, the results are similar. Table 7 shows all the relevant data.

This study reveals that the nutritional status of the investigated children does not differ between girls (*χ*² = 11.40; *p* = 0.877) and boys (*χ*² = 12.51; *p* = 0.820) according to the choice of products from the school shop.

## 4. Discussion

The importance of proper dietary behaviors in terms of health and obesity prevention among children cannot be overestimated. Our study is one of few scientific studies on the dietary habits of children in Poland. In addition, our study is distinguished by a relatively large study group from different regions of the country. Analyzing the nutrition of the children examined, it is noteworthy that less than 1/5 of children have excessive body weight, and 6.8% of children are obese. According to data from the European COSI report co-created by WHO, and assessing the nutrition of European children aged 7–9 years and their selected health behaviors, an average of 29% of boys and 27% of girls in Europe are overweight, of which, 13% of boys and 9% of girls suffer from obesity. According to this report, the percentage of disorders among Polish children resulting from excessive body weight reaches a slightly higher level than average. Namely, the problem of obesity affects 14% of Polish boys and 10% of girls [3]. In our study the problem of obesity concerned a smaller percentage of children; that is, 7.0% of girls and 6.6% of boys. According to the COSI study, 84.5% of studied children in Poland eat a first breakfast, whereas 1% do not eat that meal at all [3]. The findings from the “Anthropometry, Intake and Energy Balance” (ANIBES) study are similar. The study reported that around 85% of the Spanish population (age range 9–75) were regular breakfast consumers, although one in five adolescents were breakfast skippers [8]. Comparing the data obtained in our study regarding the first breakfast routine, they are better, as almost 92% of children eat breakfast as the first meal of the day. However, in another report evaluating eating behaviors among adolescents, the percentage of breakfast daily consumption declines to only 45%. The adolescents answered that the reason for breakfast skipping is most often a lack of time and appetite in the morning [9]. Comparable results were also obtained in other reports [10,11]. According to Cheng et al., in a study that included 426 children aged 10–14 years from four local schools in Queensland, almost 1/3 children skip breakfast at least once during the school week. The study also revealed that skipping breakfast among children was associated with a lack of perceived parental emphasis on consuming breakfast [12]. In a review by Gibney et al., it was found that healthy regular breakfast consumption is associated with improved cognitive health and nutritional status and lower plasma cholesterol levels among children and adolescents [13]. Regarding dinner frequency consumption, Stefańska et al. indicate that dinner was the most frequently consumed meal (98.4–99.2% depending on gender and age) [14]. These results are in accordance to those obtained in our study.

According to recommendations, vegetables and fruits should be dominant in the diet. Despite their availability, the daily consumption of such products is relatively low. According to the 2020 data of Statistics Poland, the monthly fruit consumption was 3.86 kg/person, whereas the vegetable consumption fluctuated around 7.72 kg/person. This gives approximately 270 g of these foods per day, and the absolute minimum for an adult is 400 g [15]. When looking at our study, just over half of respondents eat fruits or vegetables daily, and nearly 40% eat those products only several times a week. However, according to the authors, the question asked in the study has some limitations because it does not specify the number of fruits and vegetables consumed per day and does not present the amounts of fruits and vegetables separately. Our study also found an interesting correlation between an increasing BMI and the consumption of fruits and vegetables. This may mean that parents of overweight children want to eat healthier because of the already existing weight disorder. In the study by Stefańska et al., this issue is discussed in more detail. The results showed that most children aged 10–12 consume raw fruits two to three times a day, with a statistically significant predominance of girls and boys in that age group compared to children aged 12–15 years. Among them, only 3% and 26.9% of girls and boys, respectively, ate fruits two to three times per day [14]. It is observed that these data are in agreement with the study results of Harton et al.. In this study, the authors tried to assess dietary habits among preschool children aged 4–6. The results indicate that only one in four children ate fresh vegetables and one in two ate fresh fruits during the day. Regarding afternoon snacks, the consumption of fruits was claimed by the largest percentage of parents (approximately 40–45% depending on the child age) [16]. Slightly better results were obtained in another study that was carried out among preschoolers as well. It reveals that little more than 50% of respondents eat fruits and vegetables one to two times a day. An alarming result is that nearly 20% (precisely—17%) of children eat fruit and vegetables less than once a day [17]. It is worth noting that the consumption of fruit juices should not be considered as a valuable substitute for fruit due to the poor satiety effect after consumption and the high probability of drinking too much, which leads to an excessive supply of simple sugars. According to the American Academy of Pediatrics guidelines, fruit juice should not be given to children under 1 year old. However, after that period, the permissible consumption amounts of these products are as follows: 1–3 years—up to 120 mL/day, 4–6 years—up to 160 mL/day, 7–18 years—up to 200 mL [18].

Another important issue in the discussion is the frequency of the consumption of both wholemeal flour bread and fish. In our study, the obtained results in this aspect are very poor. Approximately half of the studied children consume wholemeal flour bread less than once a week or do not eat it at all. According to the recommendations, products such as wholemeal flour bread are rich in polysaccharides, and hence should be eaten every day. This ensures an adequate energy supply and even consumption per unit time. It prevents glucose level fluctuations in the blood and hence prevents the consumption of simple sugars to quickly satisfy hunger. The level of the consumption of fish—foods rich in polyunsaturated omega-3 fatty acids, which are fat-soluble vitamins—leaves much to be desired. More than half of the children consume such products once a week, whereas approximately 30% consume them even less frequently. The results obtained in this regard are similar to those of other reports, which also highlighted the under-consumption of wholemeal flour bread [19,20,21]. In Stefańska’s study [14], this kind of bread is the preferred choice for half as many girls and boys as light bread. Wholemeal flour bread was consumed every day by approximately 11–15% of children aged 10–12, whereas only approximately ¼ of children consume this product several times a day. In the largest group of children, approximately 40%, consume wholemeal flour bread less than 2–3 times a week [14]. This is slightly better than the percentage obtained in our study, but is still insufficient. Furthermore, according to cited study, fish is consumed by 70% of children less than two to three times per week [14]. According to the obtained data from Fernandez et al. in a Spanish study, 12% of studied children rejected eating fish at all. Only 2% of children ate fish daily. The most commonly consumed food by Spanish school children were “pasta” and rice, soft drinks, juices and fruits (bananas, apples and oranges), cakes, tomato, snacks and fast food [22].

The main beverage for both children and adults should be water. The Guidelines for Nutrition Standards in Poland recommend that children aged 6–9 should drink approximately 1600–1750 mL of water per day, which makes approximately six to seven glasses of water [23]. It is important to know that pure water provides the best hydration for the body, i.e., 100 g of water consumed means 100 mL of functional water obtained. The addition of 10 g of glucose reduces the functional water content to only 60 mL [24]. Compared to other European countries, the percentage share of various beverages consumed by children during the day in Poland is to the disadvantage of pure water. In Poland, among children aged 7–9, only 13% of the total fluid consumption is pure water, whereas, in other countries, this percentage increases to approximately 25–43% depending on the age of children [25]. A large study from 2015 focused on the quantity of liquids, including water, consumed by children living in 13 different countries on different continents. The study group was divided by age, namely, the first group included children aged 4–9, whereas the second group included children aged 10–17. The results showed that, among all of the children studied, water was the most common drink and was consumed at an average of 738 ± 567 mL/day. Poland also took part in the study, but the results obtained in our country are unsatisfactory and significantly different from other countries surveyed. Namely, according to the received data, the daily water consumption in Poland by children aged 4–9 is 17% for both girls and boys. In the group of older children, this percentage is slightly higher, i.e., 23% of girls and 20% of boys. According to the results of this study, the largest group of children in Poland, i.e., approximately one-third, consumed hot beverages during the day [26]. In our survey, according to the respondents, water was the most common drink among children. Almost ¾ of children drink water every day, but there is still ¼ of children who do not drink water every day. As many as 15.5% of children drink water only several times a week, which indicates that this group of children is much more likely to drink other liquids to quench their thirst. In addition, the popularity of buying water from the school cafeteria is low and is reported by only 4.5% of children. An interesting fact is that children with a higher BMI drink more water. This may mean that children who are overweight and/or obese are making a more conscious effort to avoid unnecessary calories in the form of sugary drinks in favor of water. However, this only applies to children with potentially pre-existing weight disorders.

In the study by Kwiecień et al. [27], water consumption among 1st and 3rd-grade elementary school students was more popular than in our study, with 78 and 83% of children in this age group drinking water daily, respectively. Research has shown the importance of consuming water instead of other sugary drinks in the prevention and treatment of obesity. The results of a randomized study conducted in Germany proved that the introduction of one glass of water per day to the diet of elementary school children reduces the risk of being overweight in 31% of the students studied. The observation period was one school year [28]. When analyzing the consumption of sweets and sugary drinks by children, it can be seen that the consumption of those products is still high. The results show that sweets and sugary drinks are consumed daily or several times a week by a total of approximately half of the children studied. Additionally, “something sweet” was the most popular product in the school cafeteria. Other studies support those reports [29,30]. A meta-analysis dating back to 2017 showed that children are much more likely to choose high-calorie foods, such as candy, pizza and French fries, while watching TV. At the same time, significantly less fruit and vegetables are consumed [31]. One of the more recent Polish studies focuses on assessing dietary habits during the COVID-19 lockdown. The study shows that, during the pandemic, the number of meals consumed during the day increased significantly, whereas the number of snacks consumed between meals increased by as much as 5%. Sweets were among the products whose consumption also increased during that time. On the contrary, the rate of the consumption of fast food meals, instant products and energy drinks decreased. In addition, 2/3 of respondents noticed changes in their body weight during lockdown. Even though the studied group were adults, according to the authors, the results of the study may largely translate to the pediatric population due to the closure of school facilities during that period [32]. One of the newest articles also emphasizes the emotional and physiological implications of COVID-19 pandemic among children. The authors also review disturbances in sleep routines in the pandemic period [33].

The final issue the authors are addressing is the role of proper nutrition education. It is important for both parents and children. Parents, by promoting proper dietary habits, will teach their children to perpetuate those habits. Additionally, according to research, home is a major determinant of children’s behaviors, including eating. Moreover, many poor dietary habits can reflect the parents’ lack of knowledge on the subject. By removing this factor, we have a chance for a better primary prevention of weight-related diseases, and thus better health. In 2019, there was a study published whose aim was to assess selected eating behaviors before and after the implementation of a 6-month nutrition education program among children aged 12. The research tool was a survey questionnaire conducted twice (before and after nutrition education) that included questions about selected dietary habits. The results showed that, in the educated group, there was an increase in the frequency of eating breakfast and lunch in the second assessment. The frequency of the consumption of milk and dairy products, fish and vegetables also increased in this group of children. The study also evaluated a decrease in the percentage of children consuming sugary drinks. The final evaluation also showed an increase in the proportion of both girls and boys with normal weight [34]. Results obtained in the Dutch study performed with a much wider group of respondents also showed a positive relationship between parental nutrition education and the acquisition of healthier dietary habits [35].

Our study also has some limitations. Due to the questionnaire-based nature of the study and the fact that parents entered their child’s basic anthropometric measurements into the questionnaire themselves, the obtained distribution of the nutritional status of the examined children may differ from the real one, i.e., according to the authors of the present study, the percentage of children having excessive body weight may be underestimated. Moreover, questions concerning the frequency of the consumption of some food products did not specify the exact amount consumed per serving. The same is true for the frequency of the consumption of basic meals of the day, i.e., breakfast, dinner and supper. In particular, the analysis of the frequency of the supper intake poses difficulties in interpretation due to the lack of data concerning the products eaten as the last meal of the day and their quantities.

As for the survey methodology itself, the authors point out the poor response rate of 20.2%. According to the authors, this is due to several reasons. Firstly, the authors were not always present when the questionnaires were distributed at school, mainly during meetings with parents. Some of the parents did not fill out the questionnaires at those meetings and took them home. In most cases, the questionnaire was handed back to the teacher by the child, who often did not return it. Moreover, according to a large group of parents, the questionnaire was very difficult and contained too many questions. Some parents could feel ashamed of it. These facts may have been major reasons for the poor response rate obtained in the study.

## 5. Conclusions

The dietary habits of the investigated children are largely abnormal. First of all, there is a low intake of select types of products, such as wholemeal flour bread, fish and dairy products. According to the authors, this may be due to several reasons. White bread is still much more available in retail shops and supermarkets than multi-grain bread. The price of a given product is also a determining factor, as both good quality wholemeal flour bread and fish have a relatively high price. In addition, it has been “accepted” in our culture that you only eat fish once a week.

This article also presents a tendency to consume healthier products by children with an accompanying body mass disorder. This may indicate a growing awareness of the importance of a healthy diet. However, the issue it to keep both the diet and lifestyle healthy in order to prevent body mass disorders appearing. This is particularly relevant among the children population. Another very important aspect that the authors of this paper want to emphasize is the invaluable role of a proper nutrition education. These nutrition mistakes can (but do not have to) reflect the level of the parents’ knowledge about the principles of nutrition.

In conclusion, the central role of policies and comprehensive actions involving both children and their parents should be applied to better dietary habits within all family members. Then, the changes in nutrition are much more permanent. There is a need to implement some preventive activities within as many areas of a child’s and parent’s life.

## Figures and Tables

**Figure 1 ijerph-19-00953-f001:**
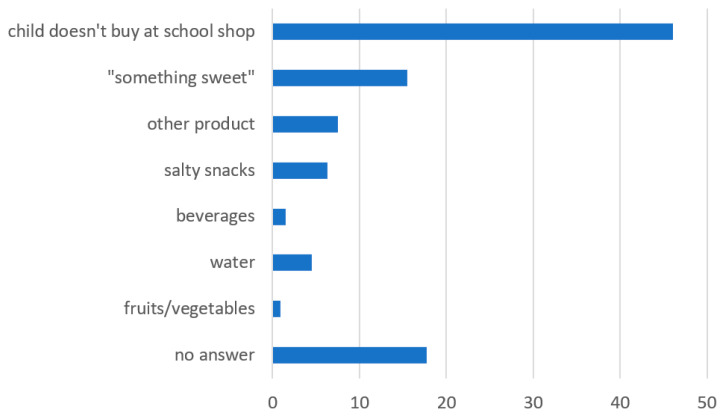
Products most willingly purchased by children from the school shop.

**Table 1 ijerph-19-00953-t001:** Distribution of the nutrition status of children surveyed according to sex.

	Underweight		Normal BMI		Overweight		Obesity	
	*n*	%	*n*	%	*n*	%	*n*	%
Girls	44	9.4	357	76.1	35	7.5	33	7.0
Boys	30	6.9	320	73.2	58	13.3	29	6.6

**Table 2 ijerph-19-00953-t002:** The frequency of consumption of meals by children surveyed.

Frequency	First Breakfast (%)	Second Breakfast (%)	Dinner (%)	Supper (%)
Daily	91.7	76.6	98.2	94.7
Several times a week	6.4	16.7	0.9	3.8
Once a week	0.2	0.2	0.2	0.2
Less than once a week	0.2	0.8	0	0.2
Never	0.2	1.1	0	0
No answer	1.3	4.6	0.7	0

**Table 3 ijerph-19-00953-t003:** The correlation between BMI of boys and girls and the frequency of main meals consumption.

		BMI	
		Girls (*n* = 450)	Boys (*n* = 419)
First breakfast	Spearman’s *rho*	0.04	−0.02
	significance	0.364	0.670
Second breakfast	Spearman’s *rho*	−0.01	−0.03
	significance	0.936	0.492
Supper	Spearman’s *rho*	0.04	−0.03
	significance	0.418	0.568

**Table 4 ijerph-19-00953-t004:** The frequency of consumption of milk, water, fish, wholemeal flour bread, meat, fruits/vegetables, beverages and sweet/salty snacks by children.

Frequency	Milk (%)	Mineral Water (%)	Fish (%)	Wholemeal Flour Bread (%)	Meat (%)	Fruits/Vegetables (%)	Beverages (%)	Sweets/Salty Snacks (%)
Daily	48.9	74.3	1.2	9.9	17.1	55.1	15	14.1
Several times a week	38.4	15.5	7.6	20.6	72.9	38.1	35.9	43.6
Once a week	4.8	2.5	52.1	13.2	4.0	3.3	18.0	18.3
Less than once a week	2.8	2.7	27.6	28.4	1.0	1.4	19.6	17.5
Never	1.9	0.7	6.7	19.9	0.6	0.6	2.9	1.2
No answer	3.3	4.4	4.8	8.0	4.5	1.5	8.5	5.3

**Table 5 ijerph-19-00953-t005:** The correlations between boys’ and girls’ BMI and consumption of milk, water, drinks, fish, wholemeal flour bread, meat, sweets/salty snacks and fruits/vegetables according to child’s age.

**Frequency of Consumption:**		**Boys’ BMI**			
		**7-Year-Olds *n* = 79**	**8-Year-Olds *n* = 108**	**9-Year-Olds *n* = 104**	**10-Year-Olds *n* = 96**
Milk and other dairy products	Spearman’s *rho*	−0.16	−0.02	−0.02	0.18
	significance	0.140	0.869	0.853	0.068
Mineral water	Spearman’s *rho*	0.01	−0.05	0.01	0.01
	significance	0.958	0.630	0.962	0.941
Beverages	Spearman’s *rho*	−0.16	0.01	−0.07	0.18
	significance	0.156	0.925	0.509	0.081
Fish	Spearman’s *rho*	−0.12	−0.02	0.06	−0.05
	significance	0.306	0.793	0.510	0.599
Wholemeal flour bread	Spearman’s *rho*	0.08	−0.10	−0.08	−0.05
	significance	0.482	0.320	0.435	0.660
Meat	Spearman’s *rho*	−0.01	0.08	−0.17	−0.01
	significance	0.902	0.415	0.073	0.960
Sweets/Salty snacks	Spearman’s *rho*	0.05	0.12	−0.10	−0.03
	significance	0.671	0.197	0.277	0.743
Fruits/Vegetables	Spearman’s *rho*	−0.18	0.11	0.06	0.20
	significance	0.106	0.225	0.501	0.044
		**Girls’s BMI**			
		**7-Year-Olds *n* = 97**	**8-Year-Olds *n* = 120**	**9-Year-Olds *n* = 65**	**10-Year-Olds *n* = 90**
Milk and other dairy products	Spearman’s *rho*	−0.11	0.13	−0.4	0.11
	Significance	0.284	0.150	0.691	0.278
Mineral water	Spearman’s *rho*	0.05	−0.09	0.22	−0.04
	significance	0.647	0.348	0.013	0.734
Beverages	Spearman’s *rho*	0.01	0.04	−0.11	−0.03
	significance	0.951	0.690	0.229	0.756
Fish	Spearman’s *rho*	−0.11	0.06	0.13	0.01
	significance	0.299	0.483	0.141	0.969
Wholemeal flour bread	Spearman’s *rho*	−0.09	−0.13	−0.01	−0.28
	significance	0.402	0.176	0.987	0.008
Meat	Spearman’s *rho*	−0.12	−0.15	−0.07	−0.25
	significance	0.226	0.090	0.430	0.013
Sweets/Salty snacks	Spearman’s *rho*	−0.06	0.02	0.01	−0.01
	significance	0.529	0.824	0.881	0.900
Fruits/Vegetables	Spearman’s *rho*	0.05	0.09	0.08	0.13
	significance	0.590	0.306	0.358	0.197

**Table 6 ijerph-19-00953-t006:** The correlation between the number of snacks consumed per day and the frequency of consumption in fast food bars with children’s BMI according to age and sex.

		**Girls’ BMI**			
		**7-Year-Olds *n* = 96**	**8-Year-Olds *n* = 121**	**9-Year-Olds *n* = 120**	**10-Year-Olds *n* = 89**
Number of snacks consumed per day	Spearman’s *rho*	0.17	−0.10	−0.08	−0.03
	significance	0.091	0.295	0.414	0.790
Frequency of consuming products in fast food bars	Spearman’s *rho*	0.13	0.03	−0.09	−0.14
	significance	0.192	0.718	0.308	0.178
		**Boys’ BMI**			
		**7-Year-Olds *n* = 80**	**8-Year-Olds *n* = 114**	**9-Year-Olds *n* = 110**	**10-Year-Olds *n* = 101**
Number of snacks consumed per day	Spearman’s *rho*	−0.01	−0.09	0.08	−0.13
	significance	0.941	0.371	0.400	0.185
Frequency of consuming products in fast food bars	Spearman’s *rho*	−0.15	0.06	−0.12	0.13
	significance	0.169	0.526	0.183	0.190

**Table 7 ijerph-19-00953-t007:** Products purchased from the school shop by girls according to age and nutritional status.

		Underweight		Normal BMI		Overweight		Obesity	
		Girls	Boys	Girls	Boys	Girls	Boys	Girls	Boys
		(%)		(%)		(%)		(%)	
6-year-olds	Something sweet	-	-	-	50.0	50.0	100.0	-	-
	Does not buy	-	-	60.0	50.0	50.0	-	-	-
	Beverages	-	-	20.0	-	-	-	-	-
	Other	-	-	20.0	-	-	-	-	-
7-year-olds	Something sweet	-	11.1	16.2	12.0	22.2	22.2	-	14.3
	Does not buy	80.0	66.7	63.2	70.0	66.7	55.6	50.0	57.1
	Salty snacks	-	-	5.9	4.0	-	11.1	25.0	-
	Beverages	10.0	11.1	1.5	2.0	-	-	-	-
	Mineral water	10.0	-	4.4	6.0	-	-	12.5	14.3
	Fruits/vegetables	-	-	1.5	-	-	-	-	-
	Other	-	11.1	7.4	6.0	11.1	11.1	12.5	14.3
8-year-olds	Something sweet	23.1	33.3	15.1	21.4	20.0	22.2	20.0	14.3
	Salty snacks	-	16.7	8.1	7.1	-	5.6	-	-
	Beverages	53.8	-	1.2	1.4	-	-	-	14.3
	Mineral water	-	33.3	4.7	5.7	20.0	5.6	-	-
	Fruits/vegetables	-	-	1.2	-	-	-	-	-
	Does not buy	-	-	58.1	50.0	60.0	55.6	80.0	57.1
	Other	23.1	-	11.6	14.3	-	11.1	-	14.3
9-year-olds	Something sweet	14.3	80.0	14.7	24.0	33.3	28.6	10.0	20.0
	Salty snacks	14.3	-	8.0	12.0	22.2	7.1	-	20.0
	Mineral water	-	20.0	1.3	6.7	11.1	14.3	-	20.0
	Does not buy	28.6	-	65.3	45.3	22.2	35.7	80.0	40.0
	Other	42.9	-	10.7	6.7	11.1	14.3	-	-
	Fruits/vegetables	-	-	-	4.0	-	-	10.0	-
	Beverages	-	-	-	1.3	-	-	-	-
10-year-olds	Something sweet	44.4	20.0	25.9	10.9	28.6	22.2	20.0	-
	Salty snacks	22.2	-	13.0	10.9	-	-	-	33.3
	Does not buy	33.3	80.0	40.7	59.4	57.1	36.4	80.0	66.7
	Beverages	-	-	1.9	3.1	-	11.1	-	-
	Mineral water	-	-	6.7	6.3	-	-	-	-
	Fruits/vegetables	-	-	1.9	1.6	-	-	-	-
	Other	-	-	5.3	7.8	14.3	22.2	-	-

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
