# Peer review of "The Analysis of the Nutritional Status and Dietary Habits among Children Aged 6–10 Years Old Attending Primary Schools in Poland"

_ijerph, 2022, doi:10.3390/ijerph19020953_

Round 1

Reviewer 1 Report

In general, the paper shows the nutrition data obtained in children and the authors seek to explain them in terms of BMI and types of food consumed. It is not an unprecedented survey. But, it can be a reference for adopting strategies that minimize the problems of overweight, which is important for the region.

The introduction is adequately described. Problems start to appear in the following items:

Material and Methods - This item needs to be completely rewritten. Leave the information in a logical sequence so that the reader is not in doubt. All changes are marked in the text.

Results - this item also requires some changes: a) the tables could be organized so that the data obtained, for example, for boys and girls, by age, could be easily visualized and compared; b) the figures are too big the figures are too large and the use of "bars" could facilitate the visualization and comparison of results between boys and girls; c) in this item, we identify the importance of describing the items in the instrument/questionnaire in terms of material and methods. There is the use of vague terms such as "several", "something", "beverage", "first and second breakfast items", several times", among others.

Discussion - I suggest that in the discussion the data is not repeated. better to write: comparing the data obtained, regarding the consumption of fruits, and other studies, it is observed that they are in agreement... they are not in agreement. It is the interpretation of data, their impact on health and comparison with other studies. I suggest rewriting. I also suggest removing what is marked at the beginning of the item until the middle of line 268.

Author Response

Dear Reviewer,

Thank You very much for reviewing my manuscript and for all Your feedback on it.

I have tried to improve the paper according to the marked annotations within the text and the comments.

Photo 1. has been removed and I have included a brief description regarding the content of this figure within the text.

The results have been slightly modified, i.e. I have tried to make them more systematic, as well as to improve the graphical appearance and clarity of the tables describing the results. Nevertheless, tables are an inseparable part of presented results, performed correlations and other frequency analyses so dividing them into several smaller ones or removing some of them is not possible from the methodology point of view.

As recommended, I have included extracts from the discussion, which I hope will improve the clarity of the text and make it a comparative discussion of the results. 

Once again, I would like to thank You very much for the opportunity to participate in the review process and I hope that the corrections made (along with language correction by a native speaker) could contribute to the possibility of publication of this article in the Journal.

Best regards,

Reviewer 2 Report

I recommend proofreading of English language by native speaker. Paper will be much more clear after English proofreading.

Figure 1 should be deleted because these are well-known WHO criteria for childhood obesity. A reference can be made, but it is not necessary to put the whole picture.

The discussion was written in a very popular way. It is unnecessary to mention Hippocrates in scientific paper or claims that proper nutrition prevents obesity, because these are well-known claims and quotations.

Author Response

Dear Reviewer, 
Thank You very much for Your review of our manuscript and all your feedback regarding the substantive content of the paper. As recommended, I have tried to make appropriate corrections in the paper (within all the chapters, including mentioned in Your Opinion Discussion too).

The paper was also reviewed by a native speaker (American English). 

Once again, thank You very much for the opportunity to participate in the review process and I hope that the corrections made could contribute to publication of this article in the Journal. 

Best regards,

Reviewer 3 Report

Dear Authors, 
Thank you for the opportunity to read your article. It raises the important issue of diet and eating habits in the developmental age population. This issue is important in the context of increasingly common health problems, including obesity in young children.
I have several comments about the manuscript:
1 - the objective is not well formulated. Previous knowledge proves the relationship between nutritional status and dietary behavior. You want to prove a thesis that has already been proven. This is a tautology. Besides, it is never the case that only one factor affects nutritional status. 
2 - In the Introduction there is only one sentence about children from Poland. Why do we focus on this age group? 
3 - Material and Methods
Please organize this section, e.g. Study Design, Setting, Data Collection, Data Analysis.
It would be good to make the description of the research more detailed with information on when and where the parents completed the questionnaires - at school, at home, how they returned the completed questionnaires, to whom etc.?
4 - the graphs should be removed as they repeat content of the description, they do not bring anything new. 
What is the purpose of placing such a large and prominent graph 1? 
5 - the results should be shorter, more synthetically presented. Please rethink the structure of the tables, they are unreadable, e.g. 6. it is called "The correlations between boys' and girls' BMI and consumption of milk, water, drinks, fish, brown bread, meat, sweets/salty snacks and fruit/vegetables" - there is gender, age, frequency of consumption of products and no BMI
6 - limitations are missing 
7 - Conclusions do not match the purpose of the study, do not show what is new in the study, how it can be used
Where did the information about knowledge, mass media come from? This was not the purpose of the study. 

Author Response

Dear Reviewer,

Thank you very much for your review of our manuscript and all your feedback regarding the substantive content of the paper. As recommended, I have tried to make appropriate corrections in the paper. The paper was also reviewed by a native speaker (American English).

As far as the introduction is concerned, I have decided that the basic statistical data on the nutritional status of Polish children will be discussed in the 'Discussion' chapter. In my opinion and that of the other co-authors this will improve the clarity of the paper, and in the introduction I have focused on presenting this problem globally.

The chapter 'Methodology' and 'Results' have been almost completely reformatted ('Methodology') and modified ('Results') in order to make the text easier to read. Thank you for this attention once again.

As for the Tables in the 'Results' section, which are an integral part of the presented results, frequency and correlation analyses, I have decided to slightly reformat them, but in a rather delicate way, in order to keep all the necessary data for the analysis, while preserving the correctness of the methodology. With regard to the tables containing the analysis of correlations of frequency of consumption of given food products with BMI of girls/boys depending on age, it is impossible to include in the table each correlated value of BMI of each child with frequency of consumption of a given product. This is pure statistical analysis, and the table shows the results of this correlation done with Spearman's rho index and statistical significance.

Again, thank you very much for the opportunity to participate in the review process and I hope that the corrections made could contribute to make a chance to publish this article in the Journal.

Best regards,

Round 2

Reviewer 1 Report

Authors can further improve data presentation.

Author Response

Dear Reviewer,

Thank You once more for the review process. I am truly honored to have a chance to take part in the revision process to the Journal.

Referring to the remarks, In my opinion creating together Table 2 and 3 in one could decrease legibility of these data. Firstly Table 2 illustrates frequency distribution of main daily meals consumption, not their correlation analysis with BMI which is presented in table 3. There are two tables which illustrates two separate data. Accordingly in the Table 3 there is presented sex division of studied children, which is not presented in Table 2.

Thank You once more for the remark about Table 4 and 5. According to the suggest I have incorporated data from Table 5 into Table 4. The same is with table 9 and 10. Hope that in this way my ‘Results’ will by more readable. In the main text in this Chapter I didn’t make any corrections. They are consistent and lack of some of them can make the text little chaotic (in my opinion; when You have other opinion I will try to correct the main text).

Moreover, according to other Reviewer opinion I have cancelled the Figures because all the data are contained in the main text as well.

Once more I want to thank You for Your time spent on reading and suggesting the modifications of the manuscript. I hope that You will be satisfied with that response.

I truly hope that this form of paper will be suitable for publication.

Best regards, 

Reviewer 3 Report

Dear Authors, 
you have put in a lot of work to improve the manuscript. I am glad that you took my advice. Good job!
However, I am unsatisfied because in the introduction you did not indicate the phenomenon described in Poland. Does it look the same as in other European countries? 
I don't understand the idea of including figures 1, 2 and 3. You can describe them and that would be enough. Figure 3 is very ugly, although this is my aesthetic impression, not a professional one.

Author Response

Dear Reviewer,

Thank You once more for the review process. I am truly honored to have a chance to take part in the revision process to the Journal.

Referring to the remarks I have cancelled the Figures because all the data are contained in the main text as well. When going to Figure 3, I have modified this Figure and in my opinion now it is more readable. Hope You will accept it.

Taking into consideration ‘Introduction’ I have added to this Chapter a sentence about Polish general situation about body mass disorders. But there is not any phenomenon in Poland when talking to body mass disorders, as I have mentioned they are in the middle or even higher than average values when compared to other European countries. However I sustain in my theory that particularly data which concern prevalence of body mass disorders have been presented in the ‘Discussion’ chapter.

Once more I want to thank You for Your time spent on reading and suggesting the modifications of the manuscript. I hope that You will be satisfied with that response.

I truly hope that this form of paper will be suitable for publication.

Best regards,